# Stochastic and Adversarial Online Learning without Hyperparameters

**Ashok Cutkosky**
Department of Computer Science
Stanford University
ashokc@cs.stanford.edu

**Kwabena Boahen**
Department of Bioengineering
Stanford University
boahen@stanford.edu

## Abstract

Most online optimization algorithms focus on one of two things: performing well in adversarial settings by adapting to unknown data parameters (such as Lipschitz constants), typically achieving $O(\sqrt{T})$ regret, or performing well in stochastic settings where they can leverage some structure in the losses (such as strong convexity), typically achieving $O(\log(T))$ regret. Algorithms that focus on the former problem hitherto achieved $O(\sqrt{T})$ in the stochastic setting rather than $O(\log(T))$. Here we introduce an online optimization algorithm that achieves $O(\log^4(T))$ regret in a wide class of stochastic settings while gracefully degrading to the optimal $O(\sqrt{T})$ regret in adversarial settings (up to logarithmic factors). Our algorithm does not require any prior knowledge about the data or tuning of parameters to achieve superior performance.

## 1 Extending Adversarial Algorithms to Stochastic Settings

The online convex optimization (OCO) paradigm [1, 2] can be used to model a large number of scenarios of interest, such as streaming problems, adversarial environments, or stochastic optimization. In brief, an OCO algorithm plays $T$ rounds of a game in which on each round the algorithm outputs a vector $w_t$ in some convex space $W$, and then receives a loss function $\ell_t : W \to \mathbb{R}$ that is convex. The algorithm's objective is to minimize *regret*, which is the total loss of all rounds relative to $w^\star$, the minimizer of $\sum_{t=1}^{T} \ell_t$ in $W$:

$$R_T(w^\star) = \sum_{t=1}^{T} \ell_t(w_t) - \ell_t(w^\star)$$

OCO algorithms typically either make as few as possible assumptions about the $\ell_t$ while attempting to perform well (adversarial settings), or assume that the $\ell_t$ have some particular structure that can be leveraged to perform much better (stochastic settings). For the adversarial setting, the minimax optimal regret is $O(BL_{\max}\sqrt{T})$, where $B$ is the diameter of $W$ and $L_{\max}$ is the maximum Lipschitz constant of the losses [3]. A wide variety of algorithms achieve this bound without prior knowledge of one or both of $B$ and $L_{\max}$ [4, 5, 6, 7], resulting in hyperparameter-free algorithms. In the stochastic setting, it was recently shown that for a class of problems (those satisfying the so-called *Bernstein condition*), one can achieve regret $O(dBL_{\max} \log(T))$ where $W \subset \mathbb{R}^d$ using the METAGRAD algorithm [8, 9]. This approach requires knowledge of the parameter $L_{\max}$.

In this paper, we extend an algorithm for the parameter-free adversarial setting [7] to the stochastic setting, achieving both optimal regret in adversarial settings as well as logarithmic regret in a wide class of stochastic settings, without needing to tune parameters. Our class of stochastic settings is those for which $\mathbb{E}[\nabla \ell_t(w_t)]$ is aligned with $w_t - w^\star$, quantified by a value $\alpha$ that increases with

increasing alignment. We call losses in this class $\alpha$-acutely convex, and show that a single quadratic lower bound on the average loss is sufficient to ensure high $\alpha$.

This paper is organized as follows. In Section 2, we provide an overview of our approach. In Section 3, we give explicit pseudo-code and prove our regret bounds for the adversarial setting. In Section 4, we formally define $\alpha$-acute convexity and prove regret bounds for the acutely convex stochastic setting. Finally, in Section 5, we give some motivating examples of acutely convex stochastic losses. Section 6 concludes the paper.

## 2 Overview of Approach

Before giving the overview, we fix some notation. We assume our domain $W$ is a closed convex subset of a Hilbert space with $0 \in W$. We write $g_t$ to be an arbitrary subgradient of $\ell_t$ at $w_t$ for all $t$, which we denote by $g_t \in \partial \ell_t(w_t)$. $L_{\max}$ is the maximum Lipschitz constant of all the $\ell_t$, and $B$ is the diameter of the space $W$. The norm $\| \cdot \|$ we use is the 2-norm: $\|w\| = \sqrt{w \cdot w}$. We observe that since each $\ell_t$ is convex, we have $R_T(w^\star) \leq \sum_{t=1}^T g_t(w_t - w^\star)$. We will make heavy use of this inequality; every regret bound we state will in fact be an upper bound on $\sum_{t=1}^T g_t(w_t - w^\star)$. Finally, we use a compressed sum notation $g_{1:t} = \sum_{t'=1}^t g_{t'}$, and we use $\tilde{O}$ to suppress logarithmic terms in big-Oh notation. All proofs omitted from the main text appear in the appendix.

Our algorithm works by trading off some performance in order to avoid knowledge of problem parameters. Prior analysis of the METAGRAD algorithm [9] showed that any algorithm guaranteeing $R_T(w^\star) = \tilde{O}\left(\sqrt{\sum_{t=1}^T (g_t \cdot (w_t - w^\star))^2}\right)$ will obtain logarithmic regret for stochastic settings satisfying the Bernstein condition. We will instead guarantee the weaker regret bound:

$$R_T(w^\star) \leq \tilde{O}\left(\sqrt{L_{\max} \sum_{t=1}^T \|g_t\| \|w_t - w^\star\|^2}\right) \tag{1}$$

which we will show in turn implies $\sqrt{T}$ regret in adversarial settings and logarithmic regret for acutely convex stochastic settings. Although (1) is weaker than the METAGRAD regret bound, we can obtain it without prior knoweldge.

In order to come up with an algorithm that achieves the bound (1), we interpret it as the square root of $\mathbb{E}[\|w - w^\star\|^2]$, where $w$ takes on value $w_t$ with probability proportional to $\|g_t\|$. This allows us to use the bias-variance decomposition to write (1) as:

$$R_T(w^\star) \leq \tilde{O}\left(\|w^\star - \overline{w}\| \sqrt{L_{\max} \|g\|_{1:T}} + \sqrt{\sum_{t=1}^T L_{\max} \|g_t\| \|w_t - \overline{w}\|^2}\right) \tag{2}$$

where $\overline{w} = \frac{\sum_{t=1}^T \|g_t\| w_t}{\|g\|_{1:T}}$. Certain algorithms for unconstrained OCO can achieve $R_T(u) = \tilde{O}(\|u\| L_{\max} \sqrt{\|g\|_{1:T}})$ simultaneously for all $u \in W$ [10, 6, 11, 7]. Thus if we knew $\overline{w}$ ahead of time, we could translate the predictions of one such algorithm by $\overline{w}$ to abtain $R_T(w^\star) \leq \tilde{O}(\|w^\star - \overline{w}\| L_{\max} \sqrt{\|g\|_{1:T}})$, the bias term of (2). We do not know $\overline{w}$, but we can estimate it over time. Errors in the estimation procedure will cause us to incur the variance term of (2). We implement this strategy by modifying FREEREX [7], an unconstrained OCO algorithm that does not require prior knowledge of any parameters.

Our modification to FREEREX is very simple: we set $w_t = \hat{w}_t + \overline{w}_{t-1}$ where $\hat{w}_t$ is the $t^{\text{th}}$ output of FREEREX, and $\overline{w}_{t-1}$ is (approximately) a weighted average of the previous vectors $w_1, \ldots, w_{t-1}$ with the weight of $w_t$ equal to $\|g_t\|$. This $\overline{w}_t$ offset can be viewed as a kind of momentum term that accelerates us towards optimal points when the losses are stochastic (which tends to cause correlated $w_t$ and therefore large offsets), but has very little effect when the losses are adversarial (which tends to cause uncorrelated $w_t$ and therefore small offsets).

# 3 FREEREXMOMENTUM

In this section, we explicitly describe and analyze our algorithm, FREEREXMOMENTUM, a modification of FREEREX. FREEREX is a Follow-the-Regularized-Leader (FTRL) algorithm, which means that for all $t$, there is some regularizer function $\psi_t$ such that $w_{t+1} = \operatorname{argmin}_W \psi_t(w) + g_{1:t} \cdot w$. Specifically, FREEREX uses $\psi_t = \frac{\sqrt{5}}{a_t \eta_t} \phi(a_t w)$, where $\phi(w) = (\|w\| + 1) \log(\|w\| + 1) - \|w\|$ and $\eta_t$ and $a_t$ are specific numbers that grow over time as specified in Algorithm 1. FREEREXMO-MENTUM's predictions are given by offsetting FREEREX's predictions $w_{t+1}$ by a momentum term $\overline{w}_t = \frac{\sum_{t'=1}^{t-1} \|g_{t'}\| w_t}{1 + \|g\|_{1:t}}$. We accomplish this by shifting the regularizers $\psi_t$ by $\overline{w}_t$, so that FREEREXMO-MENTUM is FTRL with regularizers $\psi_t(w - \overline{w}_t)$.

---

**Algorithm 1** FREEREXMOMENTUM

---

**Initialize:** $\frac{1}{\eta_0^2} \leftarrow 0$, $a_0 \leftarrow 0$, $w_1 \leftarrow 0$, $L_0 \leftarrow 0$, $\psi(w) = (\|w\| + 1) \log(\|w\| + 1) - \|w\|$
**for** $t = 1$ **to** $T$ **do**
    Play $w_t$
    Receive subgradient $g_t \in \partial \ell_t(w_t)$
    $L_t \leftarrow \max(L_{t-1}, \|g_t\|)$. // $L_t = \max_{t' \le t} \|g_t\|$
    $\frac{1}{\eta_t^2} \leftarrow \max\left(\frac{1}{\eta_{t-1}^2} + 2\|g_t\|^2, L_t \|g_{1:t}\|\right)$.
    $a_t \leftarrow \max(a_{t-1}, 1/(L_t \eta_t)^2)$
    $\overline{w}_t \leftarrow \frac{\sum_{t'=1}^{t-1} \|g_{t'}\| w_t}{1 + \|g\|_{1:t}}$
    $w_{t+1} \leftarrow \operatorname{argmin}_W \left[ \frac{\sqrt{5} \phi(a_t(w - \overline{w}_t))}{a_t \eta_t} + g_{1:t} \cdot w \right]$
**end for**

---

## 3.1 Regret Analysis

We leverage the description of FREEREXMOMENTUM in terms of shifted regularizers to prove a regret bound of the same form as (1) in four steps:

1. From [7] Theorem 13, we bound the regret by

$$R_T(w^\star) \le \sum_{t=1}^{T} g_t \cdot (w_t - w^\star)$$

$$\le \psi_T(w^\star) + \sum_{t=1}^{T} \psi_{t-1}(w_{t+1}^+) - \psi_t^+(w_{t+1}^+) + g_t \cdot (w_t - w_{t+1}^+)$$

$$+ \psi_T^+(w^\star) - \psi_T(w^\star) + \sum_{t=1}^{T-1} \psi_t^+(w_{t+2}^+) - \psi_t(w_{t+2}^+)$$

    where $\psi_t^+(w) \approx \frac{\sqrt{5} \phi(a_t(w - \overline{w}_{t-1}))}{a_t \eta_t}$ is a version of $\psi_t$ shifted by $\overline{w}_{t-1}$ instead of $\overline{w}_t$, and $w_{t+1}^+ = \operatorname{argmin}_W \psi_t^+(w) + g_{1:t} w$. This breaks the regret out into two sums, one in which we have the term $\psi_{t-1}(w_{t+1}^+) - \psi_t^+(w_{t+1}^+)$ for which the two different functions are shifted by the same amount, and one with the term $\psi_t^+(w_{t+2}^+) - \psi_t(w_{t+2}^+)$, for which the functions are shifted differently, but the arguments are the same.

2. Because $\psi_{t-1}$ and $\psi_t^+$ are shifted by the same amount, the regret analysis for FREEREX in [7] applies to the second line of the regret bound, yielding a quantity similar to $\|w^\star - \overline{w}_T\| \sqrt{L_{\max} \|g\|_{1:T}}$.

3. Next, we analyze the third line. We show that $\overline{w}_t - \overline{w}_{t-1}$ cannot be too big, and use this observation to bound the third line with a quantity similar to $\sqrt{\sum_{t=1}^{T} L_{\max} \|g_t\| (w_t - \overline{w}_T)^2}$. At this point we have enough results to prove a bound of the form (2) (see Theorem 1).

4. Finally, we perform some algebraic manipulation on the bound from the first three steps to obtain a bound of the form (1) (see Corollary 2).

The details of Steps 1-3 procedure are in the appendix, resulting in Theorem 1, stated below. Step 4 is carried out in Corollary 2, which follows.

**Theorem 1.** *Let* $\psi(w) = (\|w\|+1)\log(\|w\|+1)-\|w\|$. *Set* $L_t = \max_{t' \leq t} \|g_{t'}\|$, *and* $Q_T = 2\frac{\|g\|_{1:T}}{L_{\max}}$. *Define* $\frac{1}{\eta_t}$ *and* $a_t$ *as in the pseudo-code for* FREEREXMOMENTUM *(Algorithm 1). Then the regret of* FREEREXMOMENTUM *is bounded by:*

$$\sum_{t=1}^{T} g_t \cdot (w_t - w^\star) \leq \frac{\sqrt{5}}{Q_T \eta_T} \psi(Q_T(w^\star - \overline{w_T})) + 405 L_{\max} + 2L_{\max}B + 3\frac{L_{\max}\sqrt{2L_{\max}}}{\sqrt{1+L_1}} B \log(Ba_T + 1)$$

$$+ \sqrt{2L_{\max}\left(\|\overline{w_T}\|^2 + \sum_{t=1}^{T} \|g_t\|\|w_t - \overline{w_T}\|^2\right)\left(2 + \log\left(\frac{1+\|g\|_{1:T}}{1+\|g_1\|}\right)\right)\log(Ba_T + 1)}$$

**Corollary 2.** *Under the assumptions and notation of Theorem 1, the regret of* FREEREXMOMENTUM *is bounded by:*

$$\sum_{t=1}^{T} g_t \cdot (w_t - w^\star) \leq 2\sqrt{5}\sqrt{L_{\max}\left(\|w^\star\|^2 + \sum_{t=1}^{T} \|g_t\|\|w^\star - w_t\|^2\right)\log(2BT+1)(2 + \log(T))}$$

$$+ 405 L_{\max} + 2L_{\max}B + 3\frac{L_{\max}\sqrt{2L_{\max}}}{\sqrt{1+L_1}} B \log(2BT + 1)$$

Observe that since $w_t$ and $w^\star$ are both in $W$, $\|w^\star\|$ and $\|w_t - w^\star\|$ both are at most $B$, so that Corollary 2 implies that FREEREXMOMENTUM achieves $\tilde{O}(BL_{\max}\sqrt{T})$ regret in the worst-case, which is optimal up to logarithmic factors.

## 3.2 Efficient Implementation for $L_\infty$ Balls

A careful reader may notice that the procedure for FREEREXMOMENTUM involves computing $\text{argmin}_W \left[\frac{\sqrt{5}\psi(a_t(w-\overline{w}_t))}{a_t\eta_t} + g_{1:t} \cdot w\right]$, which may not be easy if the solution $w_{t+1}$ is on the boundary of $W$. When the $w_{t+1}$ is not on the boundary of $W$, then we have a closed-form update:

$$w_{t+1} = \overline{w}_t - \frac{g_{1:t}}{a_t\|g_{1:t}\|}\left[\exp\left(\frac{\eta_t\|g_{1:t}\|}{\sqrt{5}}\right) - 1\right] \tag{3}$$

However, when $w_{t+1}$ lies on the boundary of $W$, it is not clear how to compute it for general $W$. In this section we offer a simple strategy for the case that $W$ is an $L_\infty$ ball, $W = \prod_{i=1}^{d}[-b, b]$.

In this setting, we can use the standard trick (e.g. see [12]) of running a separate copy of FREEREXMOMENTUM for each coordinate. That is, we observe that

$$R_T(w^\star) \leq \sum_{t=1}^{T} g_t \cdot (w_t - u) = \sum_{i=1}^{d}\sum_{t=1}^{T} g_{t,i}(w_{t,i} - u_i) \tag{4}$$

so that if we run an independent online learning algorithm on each coordinate, using the coordinates of the gradients $g_{t,i}$ as losses, then the total regret is at most the sum of the individual regrets. More detailed pseudocode is given in Algorithm 2.

Coordinate-wise FREEREXMOMENTUM *is* easily implementable in time $O(d)$ per update because the FREEREXMOMENTUM update is easy to perform in one dimension: if the update (3) is outside the domain $[-b, b]$, simply set $w_{t+1}$ to $b$ or $-b$, whichever is closer to the unconstrained update. Therefore, coordinate-wise FREEREXMOMENTUM can be computed in $O(d)$ time per update.

We bound the regret of coordinate-wise FREEREXMOMENTUM using Corollary 2 and Equation (4), resulting the following Corollary.

**Algorithm 2** Coordinate-Wise FREEREXMOMENTUM

---
**Initialize:** $w_1 = 0$, $d$ copies of FREEREXMOMENTUM, $F_1,\ldots,F_d$, where each $F_i$ uses domain $W = [-b, b]$.
**for** $t = 1$ **to** $T$ **do**
  Play $w_t$, receive subgradient $g_t$.
  **for** $i = 1$ **to** $d$ **do**
    Give $g_{t,i}$ to $F_i$.
    Get $w_{t+1,i} \in [-b, b]$ from $F_i$.
  **end for**
**end for**

---

**Corollary 3.** *The regret of coordinate-wise* FREEREXMOMENTUM *is bounded by:*

$$\sum_{t=1}^{T} g_t \cdot (w_t - w^\star) \leq 2\sqrt{5} \sqrt{dL_{\max}\left(d\|w^\star\|^2 + \sum_{t=1}^{T}\|g_t\|\|w^\star - w_t\|^2\right)\log(2Tb+1)(2+\log(T))}$$

$$+ 405dL_{\max} + 2L_{\max}db + 3d\frac{L_{\max}\sqrt{2L_{\max}}}{\sqrt{1+L_1}}b\log(2bT+1)$$

## 4 Logarithmic Regret in Stochastic Problems

In this section we formally define $\alpha$-acute convexity and show that FREEREXMOMENTUM achieves logarithmic regret for $\alpha$-acutely convex losses. As a warm-up, we first consider the simplest case in which the loss functions $\ell_t$ are fixed, $\ell_t = \ell$ for all $t$. After showing logarithmic regret for this case, we will then generalize to more complicated stochastic settings.

Intuitively, an acutely convex loss function $\ell$ is one for which the gradient $g_t$ is aligned with the vector $w_t - w^\star$ where $w^\star = \text{argmin}\,\ell$, as defined below.

**Definition 4.** *A convex function $\ell$ is $\alpha$-acutely convex on a set $W$ if $\ell$ has a global minimum at some $w^\star \in W$ and for all $w \in W$, for all subgradients $g \in \partial \ell(w)$, we have*

$$g \cdot (w - w^\star) \geq \alpha\|g\|\|w - w^\star\|^2$$

With this definition in hand, we can show logarithmic regret in the case where $\ell_t = \ell$ for all $t$ for some $\alpha$-acutely convex function $\ell$. From Corollary 2, with $w^\star = \text{argmin}\,\ell$, we have

$$\sum_{t=1}^{T} g_t \cdot (w_t - w^\star) \leq \tilde{O}\left(\sqrt{L_{\max}\left(\|w^\star\|^2 + \sum_{t=1}^{T}\|g_t\|\|w^\star - w_t\|^2\right)}\right)$$

$$\leq \tilde{O}\left(\sqrt{L_{\max}\left(\|w^\star\| + \frac{1}{\alpha}\sum_{t=1}^{T} g_t \cdot (w^\star - w_t)\right)}\right) \tag{5}$$

Where the $\tilde{O}$ notation suppresses terms whose dependence on $T$ is at most $O(\log^2(T))$. Now we need a small Proposition:

**Proposition 5.** *If $a$, $b$, $c$ and $d$ are non-negative constants such that*

$$x \leq a\sqrt{bx+c} + d$$

*Then*

$$x \leq 4a^2b + 2a\sqrt{c} + 2d$$

Applying Proposition 5 to Equation (5) with $x = \sum_{t=1}^{T} g_t \cdot (w_t - w^\star)$ yields

$$R_T(u) \leq \tilde{O}\left(\frac{L_{\max}\|w^\star\|}{\alpha}\right)$$

where the $\tilde{O}$ again suppresses logarithmic terms, now with dependence on $T$ at most $O(\log^4(T))$.

Having shown that FREEREXMOMENTUM achieves logarithmic regret on fixed $\alpha$-acutely convex losses, we now generalize to stochastic losses. In order to do this we will necessarily have to make some assumptions about the process generating the stochastic losses. We encapsulate these assumptions in a stochastic version of $\alpha$-acute convexity, given below.

**Definition 6.** *Suppose for all $t$, $g_t$ is such that $\mathbb{E}[g_t|g_1, \ldots g_{t-1}] \in \partial\ell(w_t)$ for some convex function $\ell$ with minimum at $w^\star$. Then we say $g_t$ is $\alpha$-acutely convex in expectation if:*
$$\mathbb{E}[g_t] \cdot (w_t - w^\star) \geq \alpha \,\mathbb{E}[\|g_t\|\|w_t - w^\star\|^2]$$
*where all expectations are conditioned on $g_1, \ldots, g_{t-1}$.*

Using this definition, a fairly straightforward calculation gives us the following result.

**Theorem 7.** *Suppose $g_t$ is $\alpha$-acutely convex in expectation and $g_t$ is bounded $\|g_t\| \leq L_{\max}$ with probability 1. Then* FREEREXMOMENTUM *achieves expected regret:*
$$\mathbb{E}[R_T(w^\star)] \leq \tilde{O}\left(\frac{L_{\max}\|w^\star\|}{\alpha}\right)$$

*Proof.* Throughout this proof, all expectations are conditioned on prior subgradients. By Corollary 2 and Jensen's inequality we have

$$\mathbb{E}\left[\sum_{t=1}^{T} g_t \cdot (w_t - w^\star)\right] \leq \mathbb{E}\left[405L_{\max} + 2L_{\max}B + 3\frac{L_{\max}\sqrt{2L_{\max}}}{\sqrt{1+L_1}}B\log(2BT+1)\right.$$

$$\left. + 2\sqrt{5}\sqrt{L_{\max}\left(\|w^\star\|^2 + \sum_{t=1}^{T}\|g_t\|\|w^\star - w_t\|^2\right)\log(2TB+1)(2+\log(T))}\right]$$

$$\leq 405L_{\max} + 2L_{\max}B + 3\frac{L_{\max}\sqrt{2L_{\max}}}{\sqrt{\delta}}B\log(2BT+1)$$

$$+ 2\sqrt{5}\sqrt{L_{\max}\left(\|w^\star\|^2 + \sum_{t=1}^{T}\mathbb{E}[\|g_t\|\|w^\star - w_t\|^2]\right)\log(2TB+1)(2+\log(T))}$$

$$\leq 405L_{\max} + 2L_{\max}B + 3\frac{L_{\max}\sqrt{2L_{\max}}}{\sqrt{\delta}}B\log(2BT+1)$$

$$+ 2\sqrt{5}\sqrt{L_{\max}\left(\|w^\star\|^2 + \frac{1}{\alpha}\sum_{t=1}^{T}\mathbb{E}[g_t \cdot (w_t - w^\star)]\right)\log(2TB+1)(2+\log(T))}$$

Set $R = \mathbb{E}\left[\sum_{t=1}^{T} g_t(w_t - w^\star)\right]$. Then we have shown

$$R \leq 2\sqrt{5}\sqrt{L_{\max}\left(\|w^\star\|^2 + \frac{R}{\alpha}\right)\log(2TB+1)(2+\log(T))}$$

$$+ 405L_{\max} + 2L_{\max}B + 3\frac{L_{\max}\sqrt{2L_{\max}}}{\sqrt{\delta}}B\log(BT+1)$$

$$= \tilde{O}\left[\sqrt{L_{\max}\left(\|w^\star\|^2 + \frac{R}{\alpha}\right)}\right]$$

And now we use Proposition 5 to conclude:

$$\sum_{t=1}^{T}\mathbb{E}[g_t \cdot (w_t - w^\star)] = \tilde{O}\left(\frac{L_{\max}\|w^\star\|}{\alpha}\right)$$

as desired, where again $\tilde{O}$ hides at most a $O(\log^4(T))$ dependence on $T$. $\qquad\square$

Exactly the same argument with an extra factor of $d$ applies to the regret of FREEREXMOMENTUM with coordinate-wise updates.

# 5 Examples of $\alpha$-acute convexity in expectation

In this section, we show that $\alpha$-acute convexity in expectation is a condition that arises in practice, justifying the relevance of our logarithmic regret bounds. To do this, we show that a quadratic lower bound on the expected loss implies $\alpha$-acute convexity, demonstrating acutely convexity is a weaker condition than strong convexity.

**Proposition 8.** *Suppose $\mathbb{E}[g_t|g_1,\ldots,g_{t-1}] \in \partial\ell(w_t)$ for some convex $\ell$ such that for some $\mu > 0$ and $w^\star = \operatorname{argmin}\ell$, $\ell(w) - \ell(w^\star) \geq \frac{\mu}{2}\|w - w^\star\|^2$ for all $w \in W$. Suppose $\|g\| \leq L_{\max}$ with probability $1$. Then $g_t$ is $\frac{\mu}{2L_{\max}}$-acutely convex in expectation.*

*Proof.* By convexity and the hypothesis of the proposition: $\mathbb{E}[g_t] \cdot (w_t - w^\star) \geq \ell(w_t) - \ell(w^\star) \geq \frac{\mu}{2}\|w_t - w^\star\|^2 \geq \frac{\mu}{2L_{\max}}\mathbb{E}[\|g_t\|\|w_t - w^\star\|^2$ □

With Proposition 8, we see that FREEREXMOMENTUM obtains logarithmic regret for any loss that is larger than a quadratic, without requiring knowledge of the parameter $\mu$ or the Lipschitz bound $L_{\max}$. Further, this result requires only the *expected loss* $\ell = \mathbb{E}[\ell_t]$ to have a quadratic lower bound - the individual losses $\ell_t$ themselves need not do so.

The boundedness of $W$ makes it surprisingly easy to have a quadratic lower bound. Although a quadratic lower bound for a function $\ell$ is easily implied by strong convexity, the quadratic lower bound is a significantly weaker condition. For example, since $W$ has diameter $B$, $\|w\| \geq \frac{1}{B}\|w\|^2$ and so the absolute value is $\frac{1}{B}$-acutely convex, but not strongly convex. The following Proposition shows that existence of a quadratic lower bound is actually a *local* condition; so long as the expected loss $\ell$ has a quadratic lower bound in a neighborhood of $w^\star$, it must do so over the entire space $W$:

**Proposition 9.** *Supppose $\ell : W \to \mathbb{R}$ is a convex function such that $\ell(w) - \ell(w^\star) \geq \frac{\mu}{2}\|w - w^\star\|$ for all $w$ with $\|w - w^\star\| \leq r$. Then $\ell(w) - \ell(w^\star) \geq \min\left(\frac{\mu r}{2B}, \frac{\mu}{2}\right)\|w - w^\star\|^2$ for all $w \in W$.*

*Proof.* We translate by $w^\star$ to assume without loss of generality that $w^\star = 0$. Then the statement is clear for $\|w\| \leq r$. By convexity, $\ell(w) - \ell(w^\star) \geq \frac{\|w\|}{r}\left[\ell\left(\frac{rw}{\|w\|}\right) - \ell(w^\star)\right] \geq \frac{\mu r}{2}\|w\| \geq \frac{\mu r}{2B}\|w\|^2$. □

Finally, we provide a simple motivating example of an interesting problem we can solve with an $\alpha$-acutely convex loss that is not strongly convex: computing the median.

**Proposition 10.** *Let $W = [a, b]$, and $\ell_t(w) = |w - x_t|$ where each $x_t$ is drawn i.i.d. from some fixed distribution with a continuous cumulative distribution function $D$, and assume $D(x^\star) = \frac{1}{2}$. Further, suppose $|2D(w) - 1| \geq F|w - x^\star|$ for all $|w - x^\star| \leq G$. Suppose $g_t = \ell_t'(w_t)$ for $w_t \neq x_t$ and $g_t = \pm 1$ with equal probability if $w_t = x_t$. Then $g_t$ is $\min\left(\frac{FG}{b-a}, F\right)$-acutely convex in expectation.*

*Proof.* By a little calculation, $\mathbb{E}[g_t] = \ell'(w_t) = 2D(w_t) - 1$, and $\mathbb{E}[\|g_t\|] = 1$. Since $\ell'(x^\star) = 0$, $w^\star = x^\star$ (the median). For $|w_t - x^\star| \geq G$, we have $|2D(w) - 1| \geq FG$, which gives $\mathbb{E}[g_t] \cdot (w_t - w^\star) \geq \frac{FG}{b-a}\mathbb{E}[\|g_t\|](w_t - w^\star)^2$. For $|w_t - x^\star| \leq G$, we have $\mathbb{E}[g_t] \cdot (w_t - w^\star) \geq F\mathbb{E}[\|g_t\|](w_t - w^\star)^2$, so that $g_t$ is $\min\left(\frac{FG}{b-a}, F\right)$-acutely convex in expectation. □

Proposition 10 shows that we can obtain low regret for an interesting stochastic problem without curvature. The condition on the cumulative distribution function $D$ is asking only that there be positive density in a neighborhood of the median; it would be satisfied if $D'(w) \geq F$ for $|w| \leq G$.

If the expected loss $\ell$ is $\mu$-strongly convex, we can apply Proposition 8 to see that $\ell$ is $\mu/2$-aligned, and then use Theorem 7 to obtain a regret of $\tilde{O}(L_{\max}\|w^\star\|/\mu)$. This is different from the usual regret bound of $\tilde{O}(L_{\max}^2/\mu)$ obtained by Online Newton Step [13], which is due to an inefficiency in using the wearker $\alpha$-alignment condition. Instead, arguing from the regret bound of Corollary 2 directly, we can recover the optimal regret bound:

**Corollary 11.** *Suppose each $\ell_t$ is an independent random variable with $\mathbb{E}[\ell_t] = \ell$ for some $\mu$-strongly convex $\ell$ with minimum at $w^\star$. Then the expected regret of* FREEREXMOMENTUM *satisfies*

$$\mathbb{E}\left[\sum_{t=1}^{T} \ell(w_t) - \ell(w^*)\right] \leq \tilde{O}(L_{\max}^2/\mu)$$

*Where the $\tilde{O}$ hides terms that are logarithmic in $TB$.*

*Proof.* From strong-convexity, we have

$$\|w_t - w^\star\|^2 \leq \frac{2}{\mu}(\ell(w_t) - \ell(w^\star))$$

Therefore applying Corollary 2 we have

$$\mathbb{E}[R_T(w^\star)] = \mathbb{E}\left[\sum_{t=1}^{T} \ell(w_t) - \ell(w^*)\right] \leq \tilde{O}\left(\sqrt{L_{\max}^2\,\mathbb{E}[\sum_{t=1}^{T}\|w_t - w^\star\|^2]}\right)$$

$$\leq \tilde{O}(\sqrt{L_{\max}^2\,\mathbb{E}[R_T(w^\star)]})$$

So that applying Proposition 5 we obtain the desired result. □

As a result of Corollary 11, we see that FREEREXMOMENTUM obtains logarithmic regret for $\alpha$-aligned problems and also obtains the optimal (up to log factors) regret bound for $\mu$-strongly-convex problems, all without requiring any knowledge of the parameters $\alpha$ or $\mu$. This stands in contrast to prior algorithms that adapt to user-supplied curvature information such as Adaptive Gradient Descent [14] or $(\mathcal{A}, \mathcal{B})$-prod [15].

## 6 Conclusions and Open Problems

We have presented an algorithm, FREEREXMOMENTUM, that achieves both $\tilde{O}(BL_{\max}\sqrt{T})$ regret in adversarial settings and $\tilde{O}\left(\frac{L_{\max}B}{\alpha}\right)$ regret in $\alpha$-acutely convex stochastic settings without requiring any prior information about any parameters. We further showed that a quadratic lower bound on the expected loss implies acute convexity, so that while strong-convexity is sufficient for acute convexity, other important loss families such as the absolute loss may also be acutely convex. Since FREEREXMOMENTUM does not require prior information about any problem parameters, it does not require any hyperparameter tuning to be assured of good convergence. Therefore, the user need not actually know whether a particular problem is adversarial or acutely convex and stochastic, or really much of anything at all about the problem, in order to use FREEREXMOMENTUM.

There are still many interesting open questions in this area. First, we would like to find an efficient way to implement the FREEREXMOMENTUM algorithm or some variant directly, without appealing to coordinate-wise updates. This would enable us to remove the factor of $d$ we incur by using coordinate-wise updates. Second, our modification to FREEREX is extremely simple and intuitive, but our analysis makes use of some of the internal logic of FREEREX. It is possible, however, that *any* algorithm with sufficiently low regret can be modified in a similar way to achieve our results. Finally, we observe that while $\log^4(T)$ is much better than $\sqrt{T}$ asymptotically, it turns out that $\log^4(T) > \sqrt{T}$ for $T < 10^{11}$, which casts the practical relevance of our logarithmic bounds in doubt. Therefore we hope that this work serves as a starting point for either new analysis or algorithm design that further simplifies and improves regret bounds.

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
