[Supplementary Material]

# A    Theorems from Literature

In this section we reproduce here some previous theorems and notation for reference.

## A.1    Follow-the-Regularized-Leader

The Follow-the-Regularized-Leader (FTRL) framework for online optimization suggests choosing $w_{t+1}$ according to the rule:

$$w_{t+1} = \underset{W}{\mathrm{argmin}}\, g_{1:t} \cdot w + \psi_t(w)$$

where $\psi_t(w)$ is a function chosen by the algorithm called a *regularizer*. We use the following bound on the regret of FTRL, which is proved in [7]:

**Theorem 12.** *Let $g_t, \dots, g_T$ be an arbitrary sequence of subgradients. Define $g_0 = 0$ for notational convenience. Let $\psi_0, \psi_1, \dots, \psi_{T-1}$ be a sequence of regularizer functions, such that $\psi_t$ is chosen without knowledge of $g_{t+1}, \dots, g_T$. Let $\psi_1^+, \dots, \psi_T^+$ be an arbitrary sequences of regularizer functions (possibly chosen with knowledge of the full subgradient sequence). Define $w_1, \dots, w_T$ to be the outputs of FTRL with regularizers $\psi_t$: $w_{t+1} = argmin\, \psi_t(w) + g_{1:t} \cdot w$, and define $w_t^+$ for $t = 2, \dots, T+1$ by $w_{t+1}^+ = argmin\, \psi_t^+(w) + g_{1:t} \cdot w$ Then FTRL with regularizers $\psi_t$ obtains regret*

$$\sum_{t=1}^{T} g_t \cdot (w_t - u) \le \psi_T^+(u) - \psi_0(w_2^+) + \sum_{t=1}^{T} \psi_{t-1}(w_{t+1}^+) - \psi_t^+(w_{t+1}^+) + g_t \cdot (w_t - w_{t+1}^+)$$

$$+ \sum_{t=1}^{T-1} \psi_t^+(w_{t+2}^+) - \psi_t(w_{t+2}^+)$$

In the next subsection we recall the notion of an *adaptive regularizer* [7], which is a function $\psi$ whose properties make it an easy building block for FTRL regularizers $\psi_t$. The analysis of FREEREXMOMENTUM is based upon the observation that its regularizers are constructed using an adaptive regularizer.

## A.2    Adaptive Regularizers

Before defining adaptive regularizers, we briefly introduce a minor generalization of strong-convexity below:

**Definition 13.** *Let $W$ be a convex space and let $\sigma : W^2 \to \mathbb{R}$ by an arbitrary function. We say a convex function $f : W \to \mathbb{R}$ is $\sigma(\cdot, \cdot)$-strongly convex with respect to a norm $\|\cdot\|$ if for all $x, y \in W$ and $g \in \partial f(x)$ we have*

$$f(y) \ge f(x) + g \cdot (y - x) + \frac{\min(\sigma(x), \sigma(y))}{2} \|x - y\|^2$$

We will exclusively make use of the special case $\sigma(w, z) = \min(\sigma(w), \sigma(z))$, and we will write $\sigma$-strongly convex instead of $\sigma(\cdot)$-strongly convex in all cases. Next we give the definition of adaptive regularizers:

**Definition 14.** *Any differentiable function $\psi : W \to \mathbb{R}$ is called a $(\sigma, \|\cdot\|)$-adaptive regularizer if it that satisfies the following conditions:*

1. *$\psi(0) = 0$.*

2. *$\psi(x)$ is $\sigma$-strongly-convex with respect to some norm $\|\cdot\|$ for some $\sigma : W \to \mathbb{R}$ such that $\|x\| \ge \|y\|$ implies $\sigma(x) \le \sigma(y)$.*

3. *For any $C$, there exists a $B$ such that $\psi(x)\sigma(x) \ge C$ for all $\|x\| \ge B$.*

Associated to every adaptive regularizer $\psi$, we define the function $h(w) = \psi(w)\sigma(w)$, and define $h^{-1}(x) = \max_{h(x) \le x} \|x\|$

Finally, we provide a general construction that converts an adaptive regularizer into a sequence of regularizers $\psi_t$ used in FTRL (and in particular in FREEREXMOMENTUM). In the following we make use of the *dual norm* $\|\cdot\|_\star$, which is defined by $\|x\|_\star = \sup_{\|y\|=1} x \cdot y$.

**Definition 15.** *Let $\| \cdot \|$ be a norm and $\| \cdot \|_\star$ be the dual norm ($\|x\|_\star = \sup_{\|y\|=1} x \cdot y$). Let $g_1, \ldots, g_T$ be a sequence of subgradients and set $L_t = \max_{t' \leq t} \|g_t\|_\star$. Define the sequences $\frac{1}{\eta_t}$ and $a_t$ recursively by:*

$$\frac{1}{\eta_0^2} = 0$$

$$\frac{1}{\eta_t^2} = \max\left( \frac{1}{\eta_{t-1}^2} + 2\|g_t\|_\star^2, L_t \|g_{1:t}\|_\star \right)$$

$$a_1 = \frac{1}{(L_1 \eta_1)^2}$$

$$a_t = \max\left( a_{t-1}, \frac{1}{(L_t \eta_t)^2} \right)$$

*Suppose $\psi$ is a $(\sigma, \| \cdot \|)$-adaptive regularizer and $k > 0$. Let $\overline{w}_1, \ldots, \overline{w}_T$ be an arbitrary sequence of vectors. Define*

$$\psi_t(w) = \frac{k}{\eta_t a_t} \psi(a_t(w - \overline{w}_t))$$

$$w_{t+1} = \underset{w \in W}{argmin}\, \psi_t(w) + g_{1:t} \cdot w$$

In order to use Theorem 12, we'll need do define some "shadow regularizers" $\psi_t^+$, which we do below:

**Definition 16.** *Given a norm $\| \cdot \|$ and a sequence of subgradients $g_1, \ldots, g_T$, define $L_t$ and $\frac{1}{\eta_t}$ as in Definition 15, and define $L_0 = L_1$. We define $\frac{1}{\eta_t^+}$ recursively by:*

$$\frac{1}{\eta_0^+} = \frac{1}{\eta_0}$$

$$\frac{1}{(\eta_t^+)^2} = \max\left( \frac{1}{\eta_{t-1}^2} + 2\|g_t\|_\star \min(\|g_t\|_\star, L_{t-1}), L_{t-1} \|g_{1:t}\|_\star \right)$$

*Further, given a $k \geq 1$ and a non-decreasing sequence of positive numbers $a_t$, define $\psi_t^+$ by:*

$$\psi_t^+(w) = \frac{k}{\eta_t^+ a_{t-1}} \psi(a_{t-1}(w - \overline{w}_{t-1}))$$

$$w_{t+1}^+ = \underset{w \in W}{argmin}\, \psi_t^+(w) + g_{1:t} \cdot w$$

The following is the key technical Lemma from [7]. That paper does not take into account the "shifting" parameter $\overline{w}_t$ and so technically the Lemma as proven there does not apply. However, by applying the change-of-coordinates $w \mapsto w - \overline{w}_{t-1}$ we see that the "shifting" does not effect the conclusion.

**Lemma 17.** *Suppose $\psi$ is a $(\sigma, \| \cdot \|)$-adaptive regularizer and $g_1, \ldots, g_T$ is an arbitrary sequence of subgradients (possibly chosen adaptively). We use the regularizers of Definition 15. Recall that we define $h(w) = \psi(w)\sigma(w)$ and $h^{-1}(x) = argmax_{h(w) \leq x} \|w\|$. Define*

$$\sigma_{min} = \inf_{\|w\| \leq h^{-1}\left(10/k^2\right)} k\sigma(w)$$

*and*

$$D = 2 \max_t \frac{h^{-1}\left( 5\frac{L_t}{kL_{t-1}} \right)}{a_{t-1}}$$

*Then*

$$\psi_{t-1}(w_{t+1}^+) - \psi_t^+(w_{t+1}^+) + g_t(w_t - w_{t+1}^+)$$

$$\leq \begin{cases} \|g_t\|_\star \min(D, \max_t(\|w_t - w_{t+1}^+\|)) & \text{when } \|g_t\| > 2L_{t-1} \\ \frac{3\|g_t\|_\star^2 \eta_t^+}{a_{t-1}\sigma_{min}} & \text{otherwise} \end{cases}$$

We copy over four final Lemmas from [7] that we include here for reference:

**Proposition 18.** *Suppose $\psi : W \to \mathbb{R}$ is a $(\sigma, \| \cdot \|)$-adaptive regularizer. Then $\frac{\psi(aw)}{a}$ is an increasing function of $a$ for all $a > 0$ for all $w \in W$.*

**Lemma 19.** *Let $\alpha_t$ be defined by*

$$\alpha_0 = \frac{1}{(L_1\eta_1)^2}$$

$$\alpha_t = \max\left(\alpha_{t-1}, \frac{1}{(L_t\eta_t)^2}\right)$$

*Then*

$$\frac{2(\|g\|_\star)_{1:t}}{L_t} \geq a_t \geq \frac{2(\|g\|_\star^2)_{1:t}}{L_t^2}$$

**Lemma 20.**　　*1.*

$$\sum_{t|\ \|g_t\|_\star \leq 2L_{t-1}} \|g_t\|_\star^2 \eta_t^+ \leq \frac{2}{\eta_T^+}$$

　　*2. Suppose $\alpha_t$ is defined by*

$$\alpha_0 = \frac{1}{(L_1\eta_1)^2}$$

$$\alpha_t = \max\left(\alpha_{t-1}, \frac{1}{(L_t\eta_t)^2}\right)$$

　　*then*

$$\sum_{t|\ \|g_t\|_\star \leq 2L_{t-1}} \|g_t\|_\star^2 \frac{\eta_t^+}{\alpha_{t-1}} \leq 15 L_{\max}$$

**Lemma 21.** *Let $a_1, \ldots, a_M$ be a sequence of non-negative numbers such that $a_{i+1} \geq 2a_i$. Then*

$$\sum_{i=1}^{M} a_i \leq 2a_M$$

# B   Proof of Main Theorem

## B.1   Proposition 5

First, we prove the simple Proposition 5, restated below for reference:

**Proposition 5.** *If $a$, $b$, $c$ and $d$ are non-negative constants such that*

$$x \leq a\sqrt{bx+c}+d$$

*Then*

$$x \leq 4a^2 b + 2a\sqrt{c} + 2d$$

*Proof.* Suppose $x \geq 2d$. Then we have

$$\frac{x}{2} \leq a\sqrt{bx+c}$$

$$x^2 \leq 4a^2 bx + 4a^2 c$$

Now we use the quadratic formula to obtain

$$x \leq \frac{4a^2 b}{2} + \frac{\sqrt{16a^4 b^2 + 16a^2 c}}{2}$$

$$\leq 4a^2 b + 2a\sqrt{c}$$

Since we assumed $x \geq 2d$ to obtain this bound, we conclude that $x$ is at most the maximum of $4a^2 b + 2a\sqrt{c}$ and $2d$, which is bounded by their sum.  □

## B.2   Proof of Theorem 1

Our strategy is based on the observation that FREEREXMOMENTUM is FTRL with regularizers $\psi_t(w) = \frac{k}{a_t \eta_t}\phi(a_t\|w - \overline{w}_t\|)$ for $\phi(x) = (x+1)\log(x+1) - x$ and $k = \sqrt{5}$, as can be easily verified by inspection of the updates. We will derive results for the case of arbitrary $k$ and $\overline{w}_t = \frac{\sum_{t'=1}^t \|g_t\| w_t}{\delta + \|g\|_{1:t}}$ for arbitrary $\delta$, and then substitute $k = \sqrt{5}$ and $\delta = 1$ at the end to derive the bound for FREEREXMOMENTUM. We think this strategy clarifies the roles of the constants in the regret bound.

The following Theorem is nearly identical to the result in [7], but is very slightly generalized to our purposes:

**Theorem 22.** *Suppose $\psi$ is a $(\sigma, \|\cdot\|)$-adaptive regularizer and $g_1, \ldots, g_T$ is some arbitrary sequence of subgradients.*

*Set*

$$\sigma_{min} = \inf_{\|w\| \leq h^{-1}\left(10/k^2\right)} k\sigma(w)$$

$$D = \max_t \max\left(\frac{L_{t-1}^2}{(\|g\|_\star^2)_{1:t-1}} h^{-1}\left(\frac{5L_t}{k^2 L_{t-1}}\right), \|w_t - w_{t+1}^+\|\right)$$

$$Q_T = 2\frac{\|g\|_{1:T}}{L_{\max}}$$

*Then FTRL with regularizers $\psi_t$ achieves regret*

$$R_T(u) \leq \frac{k}{Q_T \eta_T} \psi(Q_T(u - \overline{w_T})) + \frac{45 L_{\max}}{\sigma_{min}} + 2L_{\max} D + \psi_T^+(u) - \psi_T(u) + \sum_{t=1}^{T-1} \psi_t^+(w_{t+2}^+) - \psi_t(w_{t+2}^+)$$

*Proof.* Using Theorem 12 and Lemma 17, our regret is bounded by

$$R_T(u) \leq \psi_T(u) + \sum_{t=1}^{T} \psi_{t-1}(w_{t+1}^+) - \psi_t^+(w_{t+1}^+) + g_t(w_t - w_{t+1}^+)$$

$$+ \psi_T^+(u) - \psi_T(u) + \sum_{t=1}^{T-1} \psi_t^+(w_{t+2}^+) - \psi_t(w_{t+2}^+)$$

$$\leq \psi_T(u) + \sum_{t=1}^{T} \psi_{t-1}(w_{t+1}^+) - \psi_t^+(w_{t+1}^+) + g_t(w_t - w_{t+1}^+)$$

$$+ \psi_T^+(u) - \psi_T(u) + \sum_{t=1}^{T} \psi_t^+(w_{t+2}^+) - \psi_t(w_{t+2}^+)$$

$$\leq \psi_T(u) + \sum_{\|g_t\|_\star \leq 2L_{t-1}} \frac{3\|g_t\|^2 \eta_t^+}{a_{t-1}\sigma_{min}} + \sum_{\|g_t\|_\star > 2L_{t-1}} \|g_t\|_\star D'$$

$$+ \psi_T^+(u) - \psi_T(u) + \sum_{t=1}^{T-1} \psi_t^+(w_{t+2}^+) - \psi_t(w_{t+2}^+)$$

where $D'$ is defined by

$$D' = 2\max_t \frac{h^{-1}\left(5\frac{L_t}{kL_{t-1}}\right)}{a_{t-1}}$$

Now we use Lemma 19 to conclude that

$$D' \leq D = \max_t \frac{L_{t-1}^2}{(\|g\|_\star^2)_{1:t-1}} h^{-1}\left(5\frac{L_t}{kL_{t-1}}\right)$$

so that we have

$$R_T(u) \leq \psi_T(u) + \sum_{\|g_t\|_\star \leq 2L_{t-1}} \frac{3\|g_t\|^2 \eta_t^+}{a_{t-1}\sigma_{min}} + \sum_{\|g_t\|_\star > 2L_{t-1}} \|g_t\|_\star D$$

$$+ \psi_T^+(u) - \psi_T(u) + \sum_{t=1}^{T} \psi_t^+(w_{t+2}^+) - \psi_t(w_{t+2}^+)$$

Now using Lemma 20 we can simplify this to

$$R_T(u) \leq \frac{k}{a_T \eta_T^+} \psi(a_T u) + \frac{45 L_{\max}}{\sigma_{min}} + \sum_{\|g_t\|_\star > 2L_{t-1}} \|g_t\|_\star D$$

$$+ \psi_T^+(u) - \psi_T(u) + \sum_{t=1}^{T} \psi_t^+(w_{t+2}^+) - \psi_t(w_{t+2}^+)$$

Next, observe that each value of $\|g_t\|_\star$ in the sum $\sum_{\|g_t\|_\star > 2L_{t-1}} \|g_t\|_\star D$ is at least twice the previous value, so that by Lemma 21 we conclude

$$R_T(u) \leq \frac{k}{a_T \eta_T^+} \psi(a_T u) + \frac{45L_{\max}}{\sigma_{\min}} + 2L_{\max} D$$

$$+ \psi_T^+(u) - \psi_T(u) + \sum_{t=1}^{T-1} \psi_t^+(w_{t+2}^+) - \psi_t(w_{t+2}^+)$$

Finally, we observe that (by Lemma 19), $a_T \leq 2\frac{\|g\|_{1:T}}{L_T} = Q_T$, which gives the first inequality in the Theorem statement. $\qquad\square$

We need the next theorem to convert $\frac{45L_{\max}}{\sigma_{\min}}$ to $405L_{\max}$:

**Lemma 23.** *Suppose* $\psi(w) = ((\|w\|+1)\log(\|w\|+1) - \|w\|)$. *Then* $\psi$ *is a* $(\frac{1}{\|\cdot\|+1}, \|\cdot\|)$*-adaptive regularizer. Using the terminology of Theorem 22, for* $k = \sqrt{5}$, $\frac{45L_{\max}}{\sigma_{\min}} \leq 405L_{\max}$.

*Proof.* The fact that $\psi$ is an adaptive regularizer is proved in [7] Proposition 9. For the second statement, we have

$$\frac{45L_{\max}}{\sigma_{\min}} = \frac{45L_{\max}}{\inf_{\|w\| \leq h^{-1}(10/k^2)} k\sigma(w)}$$

$$= \sup_{\|w\| \leq h^{-1}(10/k^2)} \frac{45L_{\max}(\|w\|+1)}{k}$$

$$= \frac{45L_{\max}(h^{-1}(10/k^2)+1)}{k}$$

Now it remains to compute an expression for $h^{-1}$. First we compute a bound on $h$:

$$h(w) = \left(\log(\|w\|+1) - \frac{\|w\|}{\|w\|+1}\right)$$

$$\geq \log(\|w\|+1) - 1$$

so that

$$h^{-1}(x/k^2) \leq \exp(x/k^2 + 1) - 1$$

Now we numerically evaluate $\frac{45L_{\max}}{\sigma_{\min}} = \frac{45L_{\max}(h^{-1}(10/k^2)+1)}{k}$ using $k = \sqrt{5}$ to conclude the desired bound. $\qquad\square$

So now we go to work to bound $\psi_T^+(u) - \psi_T(u) + \sum_{t=1}^{T-1} \psi_t^+(w_{t+2}^+) - \psi_t(w_{t+2}^+)$.

**Lemma 24.** *For any increasing sequence of numbers* $\{x_t\}$,

$$\sum_{t=1}^{T} \frac{x_t - x_{t-1}}{x_t} \leq \log\left(\frac{x_T}{x_1}\right)$$

*Proof.* By concavity of log, we have

$$\log(x_t) - \log(x_{t-1}) \geq \frac{x_t - x_{t-1}}{x_t}$$

from which the result easily follows by telescoping a sum. $\qquad\square$

**Lemma 25.** *Suppose* $\{x_t\}$ *and* $\{\sigma_t\}$ *are non-negative real numbers such that* $\sqrt{x_t}\sigma_t \geq \sqrt{x_{t-1}}\sigma_{t-1}$ *for all* $t$. *Then*

$$\sum_{t=1}^{T} \frac{(x_t - x_{t-1})\sigma_t}{\sqrt{x_t}} \leq \sqrt{x_T}\sigma_T \log\left(\frac{x_T}{x_1}\right)$$

*Proof.* We have $\sqrt{x_t}\sigma_t \leq \sqrt{x_T}\sigma_T$ so that

$$\sigma_t \leq \frac{\sqrt{x_T}\sigma_T}{\sqrt{x_t}}$$

Therefore

$$\sum_{t=1}^{T} \frac{(x_t - x_{t-1})\sigma_t}{\sqrt{x_t}} \leq \sum_{t=1}^{T} \frac{(x_t - x_{t-1})\sqrt{x_T}\sigma_T}{x_t}$$

$$\leq \sqrt{x_T}\sigma_T \log\left(\frac{x_T}{x_1}\right)$$

□

We make a suggestive definition:

**Definition 26.** *Given some $\delta > 0$,*

$$x_t = \delta + (\|g\|_\star)_{1:t}$$

$$\overline{w_t} = \frac{(\|g\|_\star w)_{1:t}}{x_t}$$

$$\sigma_t = \sqrt{\frac{\delta\|\overline{w_t}\|^2 + \sum_{t'=1}^{t} \|g_t\|_\star \|w_{t'} - \overline{w_t}\|^2}{x_t}}$$

Observe that the values of $\overline{w_t}$ given in the psuedo-code for FREEREXMOMENTUM match the values above for $\delta = 1$. We will carry through all our calculations for general $\delta$, and then substitute $\delta = 1$ at the very end to obtain our regret bound.

Consider a random vector that takes on value $w_t \neq 0$ for $t \leq T$ with probability proportional to $\|g_t\|_\star$ and value $0$ with probability proportional to $\delta + \sum_{w_t=0}^{T} \|g_t\|_\star$. Then the expectation of this vector is $\overline{w_T}$ and $\sigma_T^2$ is its variance. Thus for any vector $X$, by a standard bias-variance decomposition we have

$$\delta X^2 + \sum_{t=1}^{T} \|g_t\|_\star \|X - \overline{w_T}\|^2 = x_T(\sigma_T^2 + \|X - \overline{w_T}\|^2)$$

**Lemma 27.** *Using the definitions in Definition 26, for all $T$:*

$$\sigma_T\sqrt{x_T} - \sigma_{T-1}\sqrt{x_{T-1}} \geq \frac{\|g_T\|_\star \|w_T - \overline{w_T}\|^2}{2\sigma_T\sqrt{x_T}}$$

*Proof.*

$$\sigma_T\sqrt{x_T} = \sqrt{\delta\|\overline{w_T}\|^2 + \sum_{t=1}^{T} \|g_t\|_\star \|w_t - \overline{w_T}\|^2}$$

$$\geq \sqrt{\delta\|\overline{w_T}\|^2 + \sum_{t=1}^{T-1} \|g_t\|_\star \|w_t - \overline{w_T}\|^2 + \frac{\|g_T\|_\star \|w_T - \overline{w_T}\|^2}{2\sqrt{\delta\|\overline{w_T}\|^2 + \sum_{t=1}^{T} \|g_t\|_\star \|w_t - \overline{w_T}\|^2}}}$$

$$= \sqrt{\delta\|\overline{w_T}\|^2 + \sum_{t=1}^{T-1} \|g_t\|_\star \|w_t - \overline{w_T}\|^2 + \frac{\|g_T\|_\star \|w_T - \overline{w_T}\|^2}{2\sigma_T\sqrt{x_T}}}$$

And also we have

$$\delta\|\overline{w_T}\|^2 + \sum_{t=1}^{T-1} \|g_t\|_\star \|w_t - \overline{w_T}\|^2 = x_{T-1}(\sigma_{T-1}^2 + \|\overline{w_T} - \overline{w_{T-1}}\|^2)$$

$$\geq x_{T-1}\sigma_{T-1}^2$$

and so we can conclude the desired inequality. □

**Lemma 28.** *Again using the terms from Definition 26, we have*

$$\sum_{t=1}^{T} \frac{\|g_t\|_\star \|w_t - \overline{w_t}\|}{\sqrt{x_t}} \leq \sigma_T\sqrt{x_T}\left(2 + \log\left(\frac{x_T}{\delta + L_1}\right)\right)$$

*Proof.* From Lemma 27, we see that when $\|w_t - \overline{w_t}\| \geq \sigma_t$, we have $\frac{\|g_t\|_\star \|w_t - \overline{w_t}\|}{\sqrt{x_t}} \leq 2\sigma_t \sqrt{x_t} - 2\sigma_{t-1}\sqrt{x_{t-1}}$ so that we can write:

$$\sum_{t=1}^{T} \frac{\|g_t\|_\star \|w_t - \overline{w_t}\|}{\sqrt{x_t}} \leq 2\sigma_T\sqrt{x_T} + \sum_{t=1}^{T} \frac{\|g_t\|_\star \sigma_t}{\sqrt{x_t}}$$

Now we observe (e.g. by Lemma 27) that $\sigma_t \sqrt{x_t} \geq \sigma_{t-1}\sqrt{x_{t-1}}$ for all $t$ and that $\|g_t\|_\star = x_t - x_{t-1}$ so that applying Lemma 25 we have

$$\sum_{t=1}^{T} \frac{\|g_t\|_\star \|w_t - \overline{w_t}\|}{\sqrt{x_t}} \leq 2\sigma_T\sqrt{x_T} + \sigma_T\sqrt{x_T}\log\left(\frac{x_T}{x_1}\right)$$

as desired. $\qquad\square$

**Proposition 29.** *Let $a_1, \ldots, a_T$ be non-negative numbers. Then*

$$\sum_{t=1}^{T} \frac{a_t}{(a_{1:t})^{3/2}} \leq \frac{3}{\sqrt{a_1}} - \frac{2}{\sqrt{a_{1:T}}}$$

*Proof.* We proceed by induction. For the base case, we have

$$\sum_{t=1}^{1} \frac{a_t}{(a_{1:t})^{3/2}} = \frac{1}{\sqrt{a_1}}$$

Suppose that $\sum_{t=1}^{T} \frac{a_t^2}{(a_{1:t})^{3/2}} \leq \frac{3}{\sqrt{a_1}} - \frac{2}{\sqrt{a_{1:T}}}$.

By concavity of $-\frac{1}{\sqrt{x}}$ we have

$$\left(\frac{3}{\sqrt{a_1}} - \frac{2}{\sqrt{a_{1:T+1}}}\right) - \left(\frac{3}{\sqrt{a_1}} - \frac{2}{\sqrt{a_{1:T}}}\right) \geq \frac{a_{T+1}}{(a_{1:T+1})^{3/2}}$$

By the induction assumption we have

$$\sum_{t=1}^{T+1} \frac{a_t}{(a_{1:t})^{3/2}} \leq \frac{3}{\sqrt{a_1}} - \frac{2}{\sqrt{a_{1:T}}} + \frac{a_{T+1}}{(a_{1:T+1})^{3/2}}$$

$$\leq \frac{3}{\sqrt{a_1}} - \frac{2}{\sqrt{a_{1:T+1}}}$$

as desired. $\qquad\square$

**Lemma 30.** *Define $\overline{w_t}$ as in Definition 26. Define $M_t = \sup_{w,w' \in W} \|\nabla\psi(a_t(w - w'))\|_\star$. Then using the terminology of Definition 15, we have*

$$\psi_T^+(u) - \psi_T(u) + \sum_{t=1}^{T-1} \psi_t^+(w_{t+2}^+) - \psi_t(w_{t+2}^+) \leq \sigma_T\sqrt{2L_{\max}x_T}\left(2 + \log\left(\frac{x_T}{x_1}\right)\right)\max_t M_t$$

$$+ 3\frac{L_{\max}\sqrt{2L_{\max}}}{\sqrt{\delta + L_1}}\max_t \|\overline{w_{t-1}} - w_t\|\max_t M_t$$

*Proof.* From Proposition 18, we see that $\frac{1}{a_{t-1}}\psi(a_{t-1}x) \leq \frac{1}{a_t}\psi(a_t x)$ for all $x$. Therefore we have:

$$\psi_t^+(w_{t+2}^+) - \psi_t(w_{t+2}^+) = \frac{1}{\eta_t^+ a_{t-1}}\psi(a_{t-1}(w_{t+2}^+ - \overline{w_{t-1}})) - \frac{1}{\eta_t a_t}\psi(a_t(w_{t+2}^+ - \overline{w_t}))$$

$$\leq \frac{1}{\eta_t a_t}\psi(a_t(w_{t+2}^+ - \overline{w_{t-1}})) - \frac{1}{\eta_t a_t}\psi(a_t(w_{t+2}^+ - \overline{w_t}))$$

$$\leq \frac{1}{\eta_t a_t}\|\nabla\psi(a_t(w_{t+2}^+ - \overline{w_{t-1}}))\|_\star a_t\|\overline{w_t} - \overline{w_{t-1}}\|$$

$$\leq \frac{\|\overline{w_t} - \overline{w_{t-1}}\|}{\eta_t}\max_t \|\nabla\psi(a_t(w_{t+2}^+ - \overline{w_{t-1}}))\|_\star$$

$$\leq \|\overline{w_t} - \overline{w_{t-1}}\|\sqrt{2L_{\max}x_t}\max_t \|\nabla\psi(a_t(w_{t+2}^+ - \overline{w_{t-1}}))\|_\star$$

$$\leq \|\overline{w_t} - \overline{w_{t-1}}\|\sqrt{2L_{\max}x_t}\max_t M_t$$

Where in the last step we observe $\frac{1}{\eta_t} \leq \sqrt{2L_{\max}(\|g\|_\star)_{1:t}} \leq \sqrt{2L_{\max}x_t}$, which can be easily deduced by induction, or from Proposition 19 of [7].

The exact same argument can be used to show

$$\psi_T^+(u) - \psi_T(u) \leq \|\overline{w_T} - \overline{w_{T-1}}\|\sqrt{2L_{\max}x_T}\max_t M_t$$

Next we characterize $\overline{w_t} - \overline{w_{t-1}}$:

$$\overline{w_{t-1}} - \overline{w_t} = \overline{w_{t-1}} - \frac{(\delta + (\|g\|_\star)_{1:t-1})\overline{w_{t-1}} + \|g_t\|_\star w_t}{\delta + (\|g\|_\star)_{1:t}}$$

$$= \frac{\|g_t\|_\star}{\delta + (\|g\|_\star)_{1:t}}(\overline{w_{t-1}} - w_t)$$

We can take this calculation one step further:

$$\psi_T^+(u) - \psi_T(u) + \overline{w_{t-1}} - \overline{w_t} = \frac{\|g_t\|_\star}{\delta + (\|g\|_\star)_{1:t}}(\overline{w_{t-1}} - w_t)$$

$$= \frac{\|g_t\|_\star}{\delta + (\|g\|_\star)_{1:t}}(\overline{w_t} - w_t) + \frac{\|g_t\|_\star}{\delta + (\|g\|_\star)_{1:t}}(\overline{w_{t-1}} - \overline{w_t})$$

$$= \frac{\|g_t\|_\star}{\delta + (\|g\|_\star)_{1:t}}(\overline{w_t} - w_t) + \frac{\|g_t\|_\star^2}{(\delta + (\|g\|_\star)_{1:t})^2}(\overline{w_{t-1}} - w_t)$$

Thus we have

$$\sum_{t=1}^{T-1} \psi_t^+(w_{t+2}^+) - \psi_t(w_{t+2}^+) \leq \sum_{t=1}^{T} \|\overline{w_t} - \overline{w_{t-1}}\|\sqrt{2L_{\max}x_t}\max_t M_t$$

$$= \sum_{t=1}^{T} \frac{\sqrt{2L_{\max}}\|g_t\|_\star\|\overline{w_t} - w_t\|}{\sqrt{x_t}}\max_t M_t$$

$$\sum_{t=1}^{T} \frac{\sqrt{2L_{\max}}\|g_t\|_\star^2\|\overline{w_{t-1}} - w_t\|}{x_t^{3/2}}\max_t M_t$$

$$\leq \sigma_T\sqrt{2L_{\max}x_T}\left(2 + \log\left(\frac{x_T}{x_1}\right)\right)\max_t M_t$$

$$+ 3\frac{L_{\max}\sqrt{2L_{\max}}}{\sqrt{\delta + L_1}}\max_t\|\overline{w_{t-1}} - w_t\|\max_t M_t$$

Where we've used Proposition 29 to conclude that

$$\sum_{t=1}^{T} \frac{\|g_t\|_\star^2}{x_t^{3/2}} \leq \frac{3L_{\max}}{\sqrt{\delta + L_1}}$$

and also used Lemma 28 in the last inequality. $\square$

Now if we restrict ourselves to a bounded domain of diameter $B$ and use the regularizer $\psi(w) = (\|w\| + 1)\log(\|w\| + 1) - \|w\|$, we obtain

$$\max_t M_t \leq \log(Ba_T)$$

so that we have

$$\sum_{t=1}^{T-1} \psi_t^+(w_{t+2}^+) - \psi_t(w_{t+2}^+) \leq \sqrt{2L_{\max}\left(\delta\|\overline{w_T}\|^2 + \sum_{t=1}^{T}\|g_t\|\|w_t - \overline{w_T}\|^2\right)}\left(2 + \log\left(\frac{\delta + \|g\|_{1:T}}{\delta + \|g_1\|}\right)\right)\log(Ba_T)$$

$$+ 3\frac{L_{\max}\sqrt{2L_{\max}}}{\sqrt{\delta + L_1}}B\log(Ba_T)$$

Combining this with Theorem 22 and Lemma 23 and using $\delta = 1$ and $k = \sqrt{5}$ we have proved a regret bound on FTRL with regularizers $\psi_t = \frac{\sqrt{5}}{\eta_t}\psi(w_t - \overline{w_t})$ with $\psi = (\|w\| + 1)\log(\|w\| + 1) - \|w\|$. Recall that FREEREXMOMENTUM is precisely FTRL with these regularizers, so we have proved Theorem 1:

**Theorem 1.** *Let* $\psi(w) = (\|w\| + 1)\log(\|w\| + 1) - \|w\|$. *Set* $L_t = \max_{t' \le t} \|g_{t'}\|$, *and* $Q_T = 2\frac{\|g\|_{1:T}}{L_{\max}}$. *Define* $\frac{1}{\eta_t}$ *and* $a_t$ *as in the pseudo-code for* FREEREXMOMENTUM *(Algorithm 1). Then the regret of* FREEREX-MOMENTUM *is bounded by:*

$$\sum_{t=1}^{T} g_t \cdot (w_t - w^\star) \le \frac{\sqrt{5}}{Q_T \eta_T} \psi(Q_T(w^\star - \overline{w_T})) + 405L_{\max} + 2L_{\max}B + 3\frac{L_{\max}\sqrt{2L_{\max}}}{\sqrt{1 + L_1}}B\log(Ba_T + 1)$$

$$+ \sqrt{2L_{\max}\left(\|\overline{w_T}\|^2 + \sum_{t=1}^{T}\|g_t\|\|w_t - \overline{w_T}\|^2\right)\left(2 + \log\left(\frac{1 + \|g\|_{1:T}}{1 + \|g_1\|}\right)\right)\log(Ba_T + 1)}$$

## B.3   Proof of Corollaries 2 and 3

First we prove Corollary 2, restated below:

**Corollary 2.** *Under the assumptions and notation of Theorem 1, the regret of* FREEREXMOMENTUM *is bounded by:*

$$\sum_{t=1}^{T} g_t \cdot (w_t - w^\star) \le 2\sqrt{5}\sqrt{L_{\max}\left(\|w^\star\|^2 + \sum_{t=1}^{T}\|g_t\|\|w^\star - w_t\|^2\right)\log(2BT + 1)(2 + \log(T))}$$

$$+ 405L_{\max} + 2L_{\max}B + 3\frac{L_{\max}\sqrt{2L_{\max}}}{\sqrt{1 + L_1}}B\log(2BT + 1)$$

*Proof.*   We need the observations

$$\psi(w) \le \|w\|\log(\|w\| + 1)$$

$$\frac{1}{\eta_T} \le \sqrt{2L_{\max}(1 + \|g\|_{1:T})}$$

$$a_T \le 2T$$

Using these identities with Theorem 1 gives us

$$\sum_{t=1}^{T} g_t \cdot (w_t - w^\star) \le \sqrt{5}\sqrt{2\|w^\star - \overline{w_T}\|^2 L_{\max}(1 + \|g\|_{1:T})}\log(2BT + 1)$$

$$+ \sqrt{2L_{\max}\left(\|\overline{w_T}\|^2 + \sum_{t=1}^{T}\|g_t\|\|w_t - \overline{w_T}\|^2\right)(2 + \log(T))\log(2BT + 1)}$$

$$+ 405L_{\max} + 2L_{\max}B + 3\frac{L_{\max}\sqrt{2L_{\max}}}{\sqrt{1 + L_1}}B\log(2BT + 1)$$

Now use $\sqrt{a} + \sqrt{b} \le \sqrt{2a + 2b}$ to reach the conclusion:

$$\sum_{t=1}^{T} g_t \cdot (w_t - w^\star) \le 2\sqrt{5}\sqrt{L_{\max}\left(\|w^\star - \overline{w_T}\|^2(1 + \|g\|_{1:T}) + \|\overline{w_T}\|^2 + \sum_{t=1}^{T}\|g_t\|\|w_t - \overline{w_T}\|^2\right)}$$

$$\times \log(2BT + 1)(2 + \log(T))$$

$$+ 405L_{\max} + 2L_{\max}B + 3\frac{L_{\max}\sqrt{2L_{\max}}}{\sqrt{1 + L_1}}B\log(2BT + 1)$$

$$\le 2\sqrt{5}\sqrt{L_{\max}\left(\|w^\star\|^2 + \sum_{t=1}^{T}\|g_t\|\|w^\star - w_t\|^2\right)\log(2TB + 1)(2 + \log(T))}$$

$$+ 405L_{\max} + 2L_{\max}B + 3\frac{L_{\max}\sqrt{2L_{\max}}}{\sqrt{1 + L_1}}B\log(2BT + 1)$$

$\square$

Now we Corollary 3, again restated below:

**Corollary 3.** *The regret of coordinate-wise* FREEREXMOMENTUM *is bounded by:*

$$\sum_{t=1}^{T} g_t \cdot (w_t - w^\star) \leq 2\sqrt{5}\sqrt{dL_{\max}\left(d\|w^\star\|^2 + \sum_{t=1}^{T}\|g_t\|\|w^\star - w_t\|^2\right)\log(2Tb+1)(2+\log(T))}$$

$$+ 405dL_{\max} + 2L_{\max}db + 3d\frac{L_{\max}\sqrt{2L_{\max}}}{\sqrt{1+L_1}}b\log(2bT+1)$$

*Proof.* The Corollary follows by application of Cauchy-Schwarz inequality to Corollary 2. Recall that

$$R_T(u) \leq \sum_{t=1}^{T} g_t \cdot (w_t - u) = \sum_{i=1}^{d}\sum_{t=1}^{T} g_{t,i}(w_{t,i} - u_i)$$

So that the regret can be computed by summing the regret bound of Corollary 2 across dimensions:

$$R_T(u) \leq 2\sqrt{5}\sum_{i=1}^{d}\sqrt{L_{\max}\left((w_i^\star)^2 + \sum_{t=1}^{T}|g_{t,i}|(w_i^\star - w_{t,i})^2\right)\log(2bT+1)(2+\log(T))}$$

$$+ d405L_{\max} + 2dL_{\max}b + 3d\frac{L_{\max}\sqrt{2L_{\max}}}{\sqrt{1+L_1}}b\log(2bT+1)$$

$$\leq 2\sqrt{5}\sqrt{dL_{\max}\left(d\|w^\star\|^2 + \sum_{i=1}^{d}\sum_{t=1}^{T}|g_{t,i}|(w_i^\star - w_{t,i})^2\right)\log(2bT+1)(2+\log(T))}$$

$$+ d405L_{\max} + 2dL_{\max}b + 3d\frac{L_{\max}\sqrt{2L_{\max}}}{\sqrt{1+L_1}}b\log(2bT+1)$$

$$\leq 2\sqrt{5}\sqrt{dL_{\max}\left(d\|w^\star\|^2 + \sum_{t=1}^{T}\|g_t\|\sqrt{\sum_{i=1}^{d}(w_i^\star - w_{t,i})^4}\right)\log(2bT+1)(2+\log(T))}$$

$$+ d405L_{\max} + 2dL_{\max}b + 3d\frac{L_{\max}\sqrt{2L_{\max}}}{\sqrt{1+L_1}}b\log(2bT+1)$$

$$\leq 2\sqrt{5}\sqrt{dL_{\max}\left(d\|w^\star\|^2 + \sum_{t=1}^{T}\|g_t\|\sum_{i=1}^{d}\|w_i^\star - w_{t,i})\|^2\right)\log(2bT+1)(2+\log(T))}$$

$$+ d405L_{\max} + 2dL_{\max}b + 3d\frac{L_{\max}\sqrt{2L_{\max}}}{\sqrt{1+L_1}}b\log(2bT+1)$$

where the first inequality follows from convexity of $\sqrt{x}$, the second from Cauchy-Schwarz, and the third because $\|x\|_4^2 = \sqrt{\sum_{i=1}^{d}x_i^4} \leq \|x\|_2^2$. $\qquad\square$