[Reviews · NeurIPS 2017]

Reviewer 1



This paper introduces a parameter-free algorithm for online learning that attains non-trivial guarantees in both the stochastic and adversarial settings simultaneously. Previous work in this area includes the MetaGrad algorithm, which was not parameter-free and required knowledge of L_max, but simultaneously attained O(B L_max \sqrt{T}) regret in the adversarial setting and O(dBL_max log(T)) in the stochastic or strongly setting. The key insight by the authors is to interpret an upper bound on the regret as an expectation over the learner's output vectors (scaled by the norm of the gradient at each round), to rewrite this expression in terms of a bias and variance term, and then to shift the regularization component of an existing parameter-free algorithm by a running estimate of the previously mentioned expectation. This results in an algorithm with O(\sqrt{T} \log(T)^2 L_max B) regret. When the loss functions satisfy a certain curvature property called \alpha-acute convexity, the algorithm is able to attain a regret in O(\log(T)^4 L_max B/\alpha). This paper introduces a clever method for an interesting area of online learning, that of parameter-free algorithms that work well in multiple environments. The manuscript is well-written and the authors provide the reader with an abundance of both insight into the problem and intuition for their solution. The main concern I have for the paper is the practical utility of their algorithm. The algorithm's adversarial guarantee includes an extra log(T)^2 factor in addition to the usual terms, and, as the authors admit, the stochastic guarantee in the stochastic setting may only be compelling for extraordinarily large values of T. Still, this paper is at the very least an interesting theoretical contribution. Some technical questions: 1) Line 107: Can the authors explain how they arrived at the 2nd inequality? 2) Line 184: Should the last expression be \frac{\mu}{2} \|w - w^*\|^2? 3) Line 187: Can the authors explain how they arrived at the 2nd inequality (i.e. .... \geq \frac{\mur}{2}\|w\|)? 4) As the authors mention, for strongly convex functions, \alpha = \mu/(2 L_max). Thus, the stochastic bound will become O(\log(T)^4B/\mu). However, the standard online convex optimization guarantee for strongly convex loss functions is O(log(T)L^2/\mu). Apart from the additional log(T) terms, can the authors discuss the difference in dependence on B vs L^2? Post author response: I've read the author response and choose to maintain my score.

Reviewer 2



The paper presents a new parameter-free algorithm for online convex optimization. The algorithm guarantees sqrt(T) regret in the adversarial setting while achieving polylog(T) expected regret in certain stochastic settings, referred to as "acutely convex" problems, in which the expected gradient at every point has positive correlation with the direction to the global optimum. The algorithm proposed by the authors is a modification of the parameter-free FreeRex algorithm (Cutkosky & Boahen, COLT'17), which was analyzed only in the adversarial online setting. The algorithmic modification itself is insightful and non-trivial, and the intuition behind it is nicely explained in Sec. 2 of the paper. That being said, I was not entirely convinced by the potential merits of the new algorithm. The (poly-)logarithmic regret guarantee of the algorithm is valid only for the class of "acutely convex" stochastic problems, which is shown to contain strongly convex problems but does not seem to contain much more than that. The authors do give an example of an acutely convex stochastic problem which is not strongly convex, but it is merely a one-dimensional toy example which I found not too convincing. In the strongly convex case, however, log(T) regret can be easily obtained (in both the adversarial and stochastic settings) using simple online gradient descent. While it is true that the proposed algorithm can automatically tune itself to exploit strong convexity without manually tuning any hyperparameters, simpler algorithms with this property were already suggested in the past (e.g., Bartlett et al, NIPS 2007; Sani et al, NIPS 2014). On top of the above, the algorithm is not very practical, as it is not at all obvious how to efficiently implement it even on very simple convex domains such as the L2 or L1 unit balls. Also, the log^4(T) regret bound of the algorithm is interesting only for extremely large values of T.

Reviewer 3



The authors introduce FreeRexMomentum, an online algorithm that shows O(sqrt(T)) complexity in the adversarial setting while remaining logarithmic (log4) in the stochastic setting. The algorithm is an iterative improvement over FreeRex (a Follow-The-Regularized-Leader) approach that requires no prior information on the problem's parameters, by adding a cumulative term to the regularizer. This term includes the past predictions weighted by the corresponding loss gradient. It acts as a momentum that accelerates the learning in the adversarial case. The algorithm is a nice contribution to the recently introduced adaptive algorithms in that it unifies the stochastic and adversarial case with a single algorithm without (much) compromising the optimality in the former case. The analyses seem sound even though I cannot appreciate the extent of the contribution here since I am not knowledgeable as for the classical "tools" used in this literature. There is one worrying aspect of the algorithm however, it is on the dependency on $d$ the dimensionality of the search space. I am not sure to which extend this makes the algorithm unpractical for some problems Finally, an experimental section with some practical applications would have greatly enriched the paper, showing that the same algorithm can be applied in different settings.